# Morphological Description and Live Weight Prediction from Body Measurements of Socorro Island Merino Lambs

**DOI:** 10.3390/ani13121978

**Published:** 2023-06-14

**Authors:** Paola Estefanía Castillo, Rafael Julio Macedo, Victalina Arredondo, José Luis Zepeda, Mauricio Valencia-Posadas, Carlos Urban Haubi

**Affiliations:** 1Facultad de Medicina Veterinaria y Zootecnia, Universidad de Colima, Tecomán 28100, Mexico; fannycastillo.fc@gmail.com (P.E.C.); varredondo0@ucol.mx (V.A.); jzepeda15@ucol.mx (J.L.Z.); 2Departamento de Veterinaria y Zootecnia, División de Ciencias de la Vida, CIS, Universidad de Guanajuato, Irapuato 36824, Mexico; mauvp001@yahoo.com.mx; 3Departamento de Ciencias Veterinarias, Universidad Autónoma de Aguascalientes, Aguascalientes 20900, Mexico; drhaubi@yahoo.com

**Keywords:** feral sheep, chest girth, multiple linear regression, exponential model, prediction equations

## Abstract

**Simple Summary:**

Prediction equations allow small farmers who do not have a scale to estimate the live weight of animals from their body measurements. The aim of this study was to describe the morphology and estimate the live weight from body measurements of Socorro Island Merino lambs. A group of Socorro Island Merino lambs was recorded from birth to year for live weight, rump width, rump length, withers height, body length, cannon bone perimeter, and chest girth, width, and depth. Sex and crossbreeding with hair breeds affected weight and mainly thoracic morphology at birth and at the year of Socorro Island Merino lambs, with the greatest and smaller weights and body measurements being observed in crossbreed males and purebred females, respectively. The highly significant and positive correlation between LW and CG suggests that this variable can be used as the sole predictor in exponential and linear models and in combination with other measurements in multiple linear models to provide a good estimate of LW. Chest measurements could be used as an indirect selection criterion to improve growth traits in this breed.

**Abstract:**

The aim of this study was to describe the morphology and estimate live weight from body measurements of Socorro Island Merino lambs. A group of Socorro Island Merino lambs was recorded from birth to year for live weight, rump width, rump length, withers height, body length, cannon bone perimeter, and chest girth, width, and depth. The effect of the lamb type on body measurements and live weight was analyzed using ANOVA, Pearson’s correlation analysis was performed to estimate the relationship between body measurements and live weight, multiple linear regressions were fitted to obtain prediction equations of live weight from the body measurements and finally, chest girth was used to generate prediction equations using linear and exponential models. At birth and at year, differences were observed in body measurements, especially those related to the thoracic region, with crossbred males showing the highest values. Live weight was correlated with almost all the body measurements, with the highest coefficients observed with chest girth, chest width, and chest depth. Live weight can be accurately predicted from multiple regression equations using several body measurements, but using only chest girth (CG) as a predictor, the exponential equation W_0–365_ = 0.9142 exp^(0.0462 CG)^ showed the best accuracy.

## 1. Introduction

Socorro Island’s Merino sheep face a serious risk of extinction because their population is reduced to fewer than one hundred individuals. The ancestors of this breed lived in feral conditions in the Revillagigedo Archipelago (18°49′17″ N, 112°45′50″ W) where they were introduced by Australian settlers in 1869 and eradicated in the year 2014 for causing soil compaction and erosion, loss of vegetation, habitat destruction, landscape alterations, and species extinction [1,2] (Figure 1).

The establishment of conservation programs for native zoo-genetic resources requires the morphometric characterization of the animal through the expression of its quantitative morpho-structural characters as body measurements [3]. Information on phenotypic traits is an important component of comparative studies of development, and the study of morphometrics allows a rigorous quantitative analysis of variation in the size and shape of organisms. Additionally, morphometrics is increasingly being used in several developmental contexts since it combines the geometric concept of shape with the techniques of statistics and constitutes a powerful and flexible set of tools for analyzing morphological variation [4].

On the other hand, live weight is one of the most important productive parameters since it is used to determine livestock management such as feeding, medicine doses, matting time, slaughter time, and growth [5]. In addition, establishing the interrelationship and correlation between animal live weight and linear body measurements is very important for determining genetic potential, establishing breed standards, and improving breeding programs [6,7,8].

Conventional weighing scales are the key standard to determine the live weight of domestic animals, but the economic impossibility of small producers to acquire them has been the main justification for the development of other weight estimation methods, among which prediction equations from body measurements are one of the most used [9,10]. However, in addition, knowing the relationship between live weight and body measurements would help the study of this small and endangered sheep population and would provide relevant information for its conservation and improvement. Therefore, the objective of this research was to describe the morphology and estimate the live weight from body measurements of Socorro Island Merino lambs.

## 2. Materials and Methods

The study was carried out at the Faculty of Veterinary Medicine and Animal Husbandry of the University of Colima, located in Tecomán, Colima, Mexico. The study included all 52 Socorro Island Merino lambs—13 purebred females, 14 purebred males, 9 crossbreed females, and 16 crossbreed males—born between 2019 and 2021, which were raised under a semi-intensive production system [11]. Crossbreed lambs belonged to F_3_ and F_4_ generations from crossbreeding Socorro Island Merino sheep with hair breeds such as Pelibuey, Katahdin, and Blackbelly (Figure 2).

According to Pares [3] and Herrera and Luque [12], lambs were recorded every 90 days from birth to year for live weight (LW), rump length (RL), rump width (RW), withers height (WH), chest girth (CG), chest width (CW), chest depth (CD), cannon bone perimeter (CP), and body length (BL). Animal management complied with the standards NOM-062-ZOO-1999 and NOM-051-ZOO-1995 [13].

Initially, to evaluate the effect of the lamb type (purebred female, purebred male, crossbreed female, crossbreed male) on body measurements and live weight in each 90 day sampling period, data were analyzed using an ANOVA with the model: Y_ij_ = μ + L_i_ + Ԑ_ij;_ where Y_ij_ is the body measurement or live weight, μ is the general mean, L_i_ is the fixed effect of ith lamb type (purebred female, crossbreed female, purebred male, crossbreed male), and Ԑ_ij_ is the random error. *Post hoc* comparisons were made using Tukey’s test, and significant differences were declared at *p* < 0.05.

Subsequently, Pearson’s correlation analysis was performed to estimate the relationship between body measurements and live weight, and multiple linear regressions were fitted to obtain prediction equations of live weight from the body measurements. These correlation estimates and prediction equations were developed with the measurements obtained in each sampling period (W_0_, W_90_, W_180_, W_270_, W_365_) and a general one (W_0–365_) was generated, including the totality of measurements recorded in all periods. Finally, chest girth, which was the body measurement with the highest correlation with live weight, was used to generate prediction equations using linear: Y = a + b(x) and exponential: Y = a exp^(bx)^ models, where Y is the live weight, x is the body measurement, and a and b are parameters [14]. The models were tested for goodness of fit using the residual mean square error (MSE) and the coefficients of determination pseudo R^2^ and R^2^ for nonlinear and linear models, respectively. Statistical analyses were performed using the SPSS v.15.0 program.

## 3. Results

At birth, crossbreed males showed greater CG, CW, CD, and LW than purebred females, while purebred males and crossbreed females presented intermediate values. At 90 and 180 days, all body measurements were similar, while at 270 days, CP was greater in crossbreed males and smaller in crossbreed females, and LW was greater in crossbreed males and smaller in purebred and crossbreed females. At 365 days, CP, CD, and LW were greater in crossbreed males and smaller in purebred females; WH was greater in crossbreed males and smaller in crossbreed females; and CG was greater in crossbreed males and smaller in purebred and crossbreed females (Table 1).

The birth weight of purebred females was positively and significantly correlated with all body measurements except for CW, whereas yearling weight was only significantly associated with RW, CG, and CW. For crossbreed females, the only body measurements that were significantly correlated with birth weight were WH, CG, and CW, while yearling weight was associated with all the thoracic measurements, CP, and RW. The general correlation (W_0–365_) showed a significant correlation between all body measurements and the live weight of both, purebred and crossbred females, with CG being the measurement that in general showed the highest correlation coefficients. BL was the only measure that was significantly associated with the birth weight of purebred males, while yearling weight only showed a lack of correlation with CW. In crossbreed males, all body measurements except RL were significantly correlated with birth weight, while yearling weight was correlated with all body measurements. As in the females, the general correlation (W_0–365_) showed a significant association between all the body measurements and the weight of the lambs, with the CG being the measurement that generally showed the highest correlation coefficients (Table 2).

For crossbreed males, CG and CP, CD and CW, as well as CG and RW, were the best estimators to predict the weight at birth at 180 and 365 days, respectively. The general equation (W_0–365_) incorporated CG, CP, and CD as predictors. In the case of purebred males, the best estimators for weight at birth, at 180 days, and at one year were BL, CG, and CD, respectively. In general (W_0–365_), the weight of these lambs can be estimated with greater precision using CG and CP as predictors (Table 3).

The birth weight of crossbred females can be predicted from WH and RW, 90-day weight from CW, and 270-day weight from CG. The general equation (W_0–365_) showed the best fit when incorporating Cg, WH, and CW as predictor variables. Live weight at birth and at 180 days in purebred females can be predicted from the CG, while for yearling weight, WH was added to improve the fit of the prediction equation. The general equation (W_0–365_) showed that CW is the measure that best predicts the weight (Table 4).

Table 5 shows the prediction equations for body weight generated by the exponential and linear models from CG, which was the measure that in general showed the highest correlation with live weight and was included in the largest number of prediction equations of all lambs. These equations showed a similar fit at all ages; however, the general exponential equation (W_0–365_) showed a better predictive fit than the linear one. The relationship between live weight and CG of lambs according to the general equations (W_0–365_) of both models is shown in Figure 3.

## 4. Discussion

As shown in Table 1, Socorro Island Merino lambs do not clearly show sexual body size dimorphism during the first year of life, which is clearly expressed at later ages [15]. Recent investigations carried out with this same group of lambs showed this phenomenon in relation to their live weight, since this was only greater in males at 90 and 315 days [11]. This very research pointed out that the maturation rate of males, especially purebred ones, is slower compared to that of females, so that they take several years to reach puberty and adult weight. It should be noted that although these animals are now in a domestic environment, their evolution over the near 150 years that they were on Socorro Island was directed by natural selection, and, as observed in feral sheep, this growth pattern probably allows ewe lambs to breed with adult dominant rams rather than young subordinate males [16]. Similar results were previously found by Costa-Junior et al. [17], who observed that except for RL, all body measurements were similar between male and female Santa Ines lambs in the milk teeth stage, while sexual size dimorphism was expressed and accentuated at later ages. On the contrary, other studies showed that male lambs consistently had higher CG, CW, BL, and WH after the 75th day of age (weaning), while CD was greater from birth and throughout the study [18]. Differences in sexual chromosomes, physiological characteristics, and the endocrinal system (type and measure of hormone secretion, especially sexual hormones), explain why males are heavier than females [19]. Other studies indicate that the sexual selection hypothesis is considered a general explanation for sexual size dimorphism, whereby intense sexual selection drives the evolution of body size in the selected sex, usually males, with weaker correlated selection on body size in the other sex [20].

Similarly, crossbreeding had little effect on lamb LW and morphology, as most body measurements were similar between purebred and crossbreed lambs. Even at birth and 365 days, stages in which variation in CG, CD, CW, CP, BL, and WH was observed, these were similar between purebred and crossbred lambs within the same sex. In agreement, purebred and crossbred Santa Inês lambs did not present differences for HG, BL, and WH from birth to 154 days [21]. Likewise, WH and RL of crossbreeding Farta × Washera lambs were like those of Farta lambs, as were BL was like those of Washera lambs and HG in the three genotypes [22]. Otherwise, the crossbreeding of an Ethiopian local breed with an exotic Awassi increased the LW, BL, CG, and WH of crossbreed lambs [23]. It should be noted that the crossbreed lambs studied here are the F3 and F4 generations of crossbreeding Socorro Island Merino ewes with hair breed rams, and as it has already been pointed out, heterosis is the superiority of a cross (F1) over the mid-parent, but it reduces in later generations after F1 due to segregation and recombination losses [24].

As in the present study, several studies indicate that CG is the body measurement that shows the greatest correlation with LW [22,25,26,27,28] so consequently, it can be used to predict live weight with a predictive error less than 10% regardless of the weight, age, and sex of the lambs [29]. Contrary to this, other studies found that LW is correlated with different body measurements depending on the sex of the lambs [23]. WH and BL are other body measurements that have been found to be highly correlated with LW [30,31].

Several regression equations were developed for the different lambs and ages studied. For all lambs and in most ages, CG alone or in combination with other measures was found to be the most important in accounting for a large proportion of variation in LW [23]. In multiple linear regression analysis, the coefficient of determination R^2^ usually increases as a new variable is added to the model, but under field conditions, LW estimation using one body measurement would be preferable to a combination of measurements since some of them are more affected by the animal’s posture, making it difficult to measure such variables accurately [5,23,32]. It is recognized in different studies that CG is among the body measurements least affected by the posture of the animal and is easier to measure than others such as BL and WH [5,23,26]. Furthermore, the inclusion of many variables in the model leads to multicollinearity, a phenomenon that occurs when two or more predictors are correlated and leads to an increase in the standard errors of the coefficients, skewing the results and affecting the interpretability of the regression model [5].

The linear prediction equation W_0–365_ = −18.540 + 0.598(CG) generated using CG as the single predictor variable was more accurate than those developed for Arsi Bale lambs (R^2^ = 0.68) [26], for Batur sheep (R^2^ = 0.836) [5], and for Pelibuey sheep (R^2^ = 0.79) [14].

Recently, the utilization of nonlinear models for estimating LW in sheep based on body measurements has emerged as a viable alternative to the traditional reliance on simple or multiple linear models. The accuracy of the equation W_0–365_ = 0.9142 exp^(0.0462 CG)^ generated by the exponential model to estimate LW from CG in this study (Pseudo R^2^ = 0.96) was higher than that shown by the equation proposed to predict the LW of Pelibuey ewes, which showed a Pseudo R^2^ = 0.82 [14]. In this regard, using the exponential model, Canul-Solís et al. [33] estimated the live weight of Pelibuey ewes from hip width with a similar precision to that found in the present study. Another study showed that with small sample sizes (10–20), logistic and saturation growth models can be used to determine the LW of Morkaraman sheep using BL as a predictor, and with sample sizes bigger than 20, incomplete gamma is the most accurate model [31].

As can be seen in Table 1, at the beginning of the study (W_0_), the experimental group consisted of 52 sheep, and at the end of the study (W_365_), it only included 39 individuals. This was due to the deaths of 13 sheep from various causes, mainly starvation and secondary (hepatogenous) photosensitization as a consequence of the consumption of grasses of the genus *Brachiaria* [34].

Finally, it is important to mention that beyond the findings shown here, this study should serve to raise awareness about the importance of preserving this animal genetic resource. Socorro Island Merino sheep lived in isolation and feral conditions for more than 130 years, during which they adapted to the difficult environmental conditions of the island, enduring, among other things, scarcity of food and fresh water, hunting, and the effects of hurricanes and the eruptions of Evermann volcano [1]. Therefore, the development of strategies that take advantage of the rusticity and adaptability to extreme conditions of this breed and promote its introduction in production systems developed in harsh environments could help to increase its population and guarantee its conservation.

## 5. Conclusions

Sex and crossbreeding with hair breeds affected weight and mainly thoracic morphology at birth and at the year of Socorro Island Merino lambs, with the greatest and smaller weights and body measurements being observed in crossbreed males and purebred females, respectively. The highly significant and positive correlation between LW and CG suggests that this variable can be used as the sole predictor in exponential and linear models and in combination with other measurements in multiple linear models to provide a good estimate of LW. Chest measurements could be used as an indirect selection criterion to improve growth traits in this breed.

## Figures and Tables

**Figure 1 animals-13-01978-f001:**
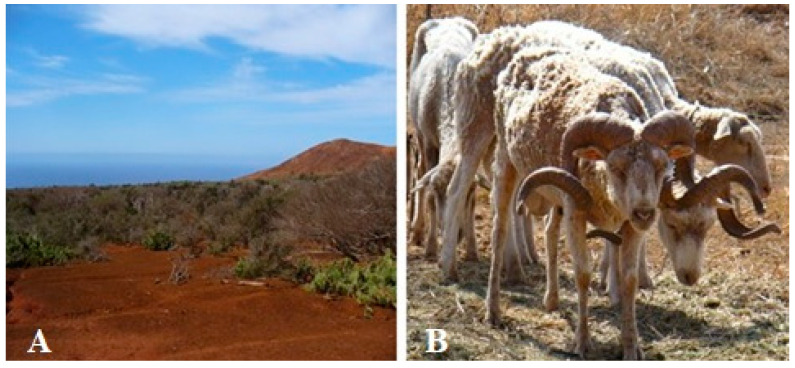
(**A**) Landscape of Socorro Island, in which the loss of vegetation caused by the grazing of the sheep can be appreciated; (**B**) Sheep captured during the rescue expedition carried out in 2006.

**Figure 2 animals-13-01978-f002:**
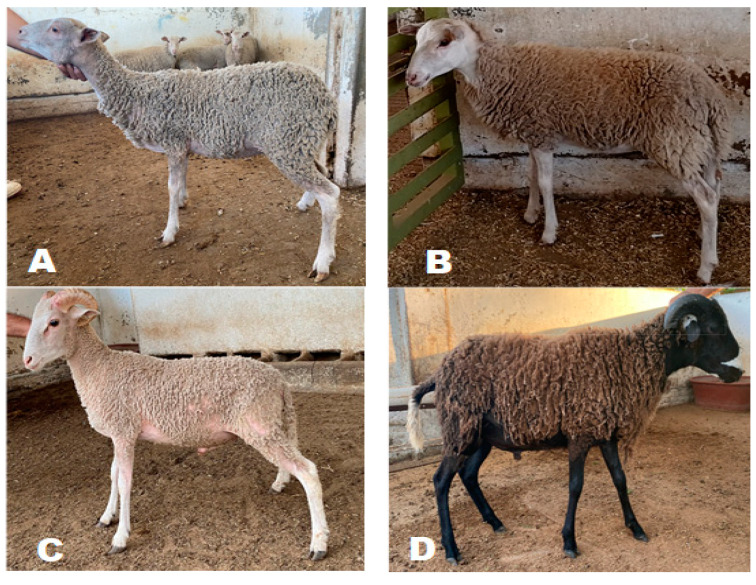
Socorro Island Merino lambs: (**A**) purebred female; (**B**) crossbreed female; (**C**) purebred male; (**D**) crossbreed male.

**Figure 3 animals-13-01978-f003:**
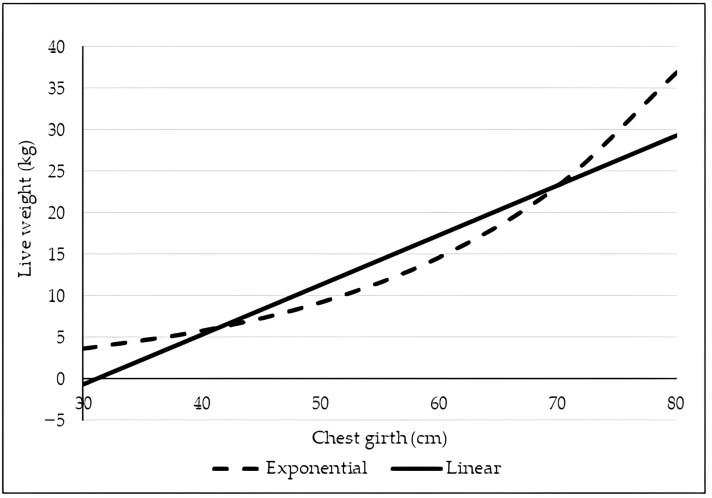
Relationship between live weight and chest girth of Socorro Island Merino lambs according to linear and exponential models.

**Table 1 animals-13-01978-t001:** Effect of type of lamb on birth to year body measurements and live weight of Socorro Island Merino lambs.

	N	RW	RL	WH	BL	CP	CG	CW	CD	LW
W_0_										
Overall	52	5.34 ± 0.08	7.61 ± 0.23	33.40 ± 0.42	25.97 ± 0.31	5.13 ± 0.07	31.58 ± 0.33	6.68 ± 0.15	11.15 ± 0.10	2.60 ± 0.07
CM	16	5.49 ± 0.14	7.93 ± 0.45	34.31 ± 0.76	26.83 ± 0.55	5.33 ± 0.12	32.99 ± 0.59 ^a^	7.33 ± 0.27 ^a^	11.88 ± 0.17 ^a^	2.91 ± 0.12 ^a^
PM	14	5.60 ± 0.16	7.60 ± 0.42	33.99 ± 0.81	26.30 ± 0.59	5.05 ± 0.13	31.34 ± 0.63 ^ab^	6.73 ± 0.29 ^ab^	10.96 ± 0.19 ^b^	2.77 ± 0.14 ^ab^
CF	9	5.12 ± 0.19	7.31 ± 0.56	32.56 ± 1.01	25.32 ± 0.74	5.09 ± 0.16	31.51 ± 0.78 ^ab^	6.66 ± 0.36 ^ab^	11.01 ± 0.23 ^b^	2.39 ± 0.18 ^ab^
PF	13	5.16 ± 0.16	7.59 ± 0.47	32.73 ± 0.84	25.42 ± 0.61	5.03 ± 0.14	30.47 ± 0.65 ^b^	5.99 ± 0.30 ^b^	10.74 ± 0.19 ^b^	2.32 ± 0.13 ^b^
W_90_										
Overall	46	9.56 ± 0.25	11.61 ± 0.34	47.20 ± 0.44	42.92 ± 0.58	5.46 ± 0.07	51.77 ± 1.24	11.39 ± 0.26	19.26 ± 0.20	10.34 ± 0.30
CM	16	9.91 ± 0.42	11.16 ± 0.58	48.96 ± 0.75	43.74 ± 0.99	5.53 ± 0.12	54.00 ± 2.09	11.95 ± 0.43	19.85 ± 0.34	11.24 ± 0.51
PM	12	10.02 ± 0.48	12.43 ± 0.67	47.46 ± 0.87	42.88 ± 1.14	5.28 ± 0.14	49.65 ± 2.42	12.00 ± 0.50	19.29 ± 0.40	10.65 ± 0.71
CF	7	9.00 ± 0.63	11.99 ± 0.88	46.24 ± 1.13	42.67 ± 1.50	5.61 ± 0.18	52.46 ± 3.17	11.49 ± 0.66	19.43 ± 0.52	9.91 ± 0.77
PF	11	9.32 ± 0.50	10.88 ± 0.70	46.13 ± 0.9	42.39 ± 1.19	5.41 ± 0.15	50.98 ± 2.52	10.14 ± 0.52	18.46 ± 0.42	9.47 ± 0.59
W_180_										
Overall	41	10.54 ± 0.26	12.00 ± 0.24	50.53 ± 0.52	45.29 ± 0.60	5.86 ± 0.07	56.80 ± 0.75	12.45 ± 0.18	20.74 ± 0.25	12.95 ± 0.52
CM	15	10.17 ± 0.43	12.15 ± 0.40	52.11 ± 0.86	46.49 ± 0.99	5.88 ± 0.11	58.51 ± 1.24	12.77 ± 0.29	21.49 ± 0.41	14.78 ± 0.86
PM	9	9.76 ± 0.53	11.85 ± 0.49	50.34 ± 1.05	43.92 ± 1.21	5.90 ± 0.14	55.41 ± 1.52	12.64 ± 0.36	20.90 ± 0.51	12.68 ± 1.06
CF	7	10.44 ± 0.63	12.59 ± 0.59	49.53 ± 1.25	45.96 ± 1.44	5.69 ± 0.17	57.24 ± 1.82	12.47 ± 0.42	20.89 ± 0.60	12.54 ± 1.27
PF	10	11.78 ± 0.56	11.42 ± 0.52	50.13 ± 1.11	44.79 ± 1.27	5.97 ± 0.15	56.02 ± 1.60	11.93 ± 0.37	19.67 ± 0.53	11.81 ± 1.12
W_270_										
Overall	41	13.20 ± 0.24	13.61 ± 0.25	55.20 ± 0.62	51.92 ± 0.83	4.46 ± 0.11	67.07 ± 1.18	14.92 ± 0.33	24.40 ± 0.39	21.33 ± 0.99
CM	15	13.15 ± 0.39	13.21 ± 0.41	57.75 ± 1.03	53.09 ± 1.37	6.84 ± 0.18 ^a^	68.85 ± 1.95	15.63 ± 0.55	25.20 ± 0.64	25.51 ± 1.63 ^a^
PM	9	13.27 ± 0.50	12.84 ± 0.53	55.61 ± 1.33	52.03 ± 1.77	6.78 ± 0.23 ^ab^	68.72 ± 2.52	16.04 ± 0.71	24.70 ± 0.82	24.07 ± 2.10 ^ab^
CF	7	12.57 ± 0.57	14.31 ± 0.60	53.70 ± 1.50	49.80 ± 2.01	5.99 ± 0.26 ^b^	64.44 ± 2.86	13.19 ± 0.81	24.04 ± 0.93	17.46 ± 2.39 ^b^
PF	10	13.80 ± 0.48	14.07 ± 0.51	53.75 ± 1.26	52.77 ± 1.68	6.24 ± 0.21 ^ab^	66.25 ± 2.39	14.83 ± 0.67	23.65 ± 0.78	18.30 ± 2.00 ^b^
W_365_										
Overall	39	13.22 ± 0.26	14.72 ± 0.31	57.35 ± 0.65	54.00 ± 0.68	6.53 ± 0.10	70.42 ± 1.00	15.43 ± 0.48	25.25 ± 0.66	25.72 ± 1.17
CM	13	13.70 ± 0.45	15.31 ± 0.54	60.96 ± 1.12 ^a^	57.09 ± 1.17	7.11 ± 0.17 ^a^	74.15 ±1.73 ^a^	16.63 ± 0.83	27.31 ± 1.15 ^a^	29.69 ± 2.02 ^a^
PM	9	13.36 ± 0.54	15.09 ± 0.65	56.39 ± 1.35 ^ab^	53.67 ± 1.41	6.87 ± 0.21 ^ab^	72.71 ± 2.09 ^ab^	15.07 ± 1.00	26.68 ± 1.38 ^ab^	29.08 ± 2.43 ^ab^
CF	7	12.57 ± 0.61	14.71 ± 0.74	53.83 ± 1.53 ^b^	52.70 ± 1.60	6.07 ± 0.24 ^bc^	67.63 ± 2.36 ^b^	14.73 ± 1.13	24.74 ± 1.57 ^ab^	22.24 ± 2.76 ^bc^
PF	10	13.27 ± 0.51	13.78 ± 0.62	58.21 ± 1.28 ^ab^	52.53 ± 1.34	6.06 ± 0.20 ^c^	67.18 ± 1.98 ^b^	15.30 ± 0.95	22.25 ± 1.31 ^b^	21.88 ± 2.31 ^c^

Note: ^abc^ Different superscript letters in the same column indicate significant differences among the data (*p* < 0.05). W_0_: Weight at birth. W_90_: Weight at 90 days (weaning). W_180_: Weight at 180 days. W_270_: Weight at 270 days. W_365_: Weight at 365 days. CM: Crossbreed male. PM: Purebred male. CF: Crossbreed female. PF: Purebred female. RW: Rump width. RL: Rump length. WH: Withers height. BL: Body length. CP: Cannon bone perimeter. CG: Chest girth. CW: Chest width. CD: Chest depth. LW: Live weight.

**Table 2 animals-13-01978-t002:** Phenotypic correlation coefficients between live weight and body measurements of Socorro Island Merino lambs.

	RW	RL	WH	BL	CP	CG	CW	CD
Crossbreed males								
W_0_	0.65 *	0.13	0.55 *	0.66 *	0.79 *	0.89 *	0.75 *	0.72 *
W_90_	0.55 *	0.50 *	0.74 *	0.84 *	0.69 *	0.88 *	0.76 *	0.90 *
W_180_	0.54 *	0.32	0.66 *	0.70 *	0.69 *	0.89 *	0.81 *	0.92 *
W_270_	0.37	0.52 *	0.86 *	0.77 *	0.95 *	0.97 *	0.97 *	0.92 *
W_365_	0.94 *	0.78 *	0.62 *	0.64 *	0.90 *	0.97 *	0.76 *	0.83 *
W_0–365_	0.89 *	0.83 *	0.91 *	0.91 *	0.91 *	0.95 *	0.95 *	0.95 *
Purebred males								
W_0_	0.43	−0.13	0.20	0.73 *	0.16	0.39	0.15	0.07
W_90_	0.50	0.65 *	0.57	0.86 *	0.80 *	0.89 *	0.84 *	0.82 *
W_180_	0.87 *	−0.18	0.60	0.72 *	0.33	0.98 *	0.90 *	0.96 *
W_270_	0.40	0.52	0.86 *	0.88 *	0.97 *	0.96 *	0.87 *	0.97 *
W_365_	0.93 *	0.67 *	0.86 *	0.91 *	0.96 *	0.94 *	0.23	0.99 *
W_0–365_	0.90 *	0.75 *	0.89 *	0.91 *	0.92 *	0.95 *	0.78 *	0.94 *
Crossbreed females								
W_0_	0.63	0.60	0.94 *	0.66	0.61	0.90 *	0.76 *	0.56
W_90_	0.55	0.20	−0.02	0.09	0.45	0.22	0.76 *	0.08
W_180_	0.84 *	0.33	0.46	0.81 *	0.50	0.85 *	0.62	0.95 *
W_270_	0.51	0.35	0.91 *	0.87 *	0.28	0.97 *	0.91 *	0.84 *
W_365_	0.90 *	0.54	0.63	0.57	0.91 *	0.97 *	0.86 *	0.86 *
W_0–365_	0.89 *	0.78 *	0.87 *	0.89 *	0.68 *	0.96 *	0.93 *	0.93 *
Purebred females								
W_0_	0.77 *	0.81 *	0.79 *	0.60 *	0.73 *	0.95 *	0.88 *	0.49
W_90_	0.41	0.16	0.65 *	0.87 *	0.07	0.75 *	0.83 *	0.81 *
W_180_	−0.25	0.13	0.02	0.64	0.16	0.85 *	0.74 *	0.71 *
W_270_	0.46	0.24	0.65 *	0.87 *	0.94 *	0.95 *	0.93 *	0.99 *
W_365_	0.77 *	−0.05	0.34	0.64	0.57	0.94 *	0.93 *	0.43
W_0–365_	0.86 *	0.75 *	0.88 *	0.91 *	0.74 *	0.94 *	0.94 *	0.83 *

Note: W_0_: Weight at birth. W_90_: Weight at 90 days (weaning). W_180_: Weight at 180 days. W_270_: Weight at 270 days. W_365_: Weight at 365 days. RW: Rump width. RL: Rump length. WH: Withers height. BL: Body length. CP: Cannon bone perimeter. CG: Chest girth. CW: Chest width. CD: Chest depth. * *p* < 0.05.

**Table 3 animals-13-01978-t003:** Multiple linear regression equations to estimate the live weight of crossbreed and purebred males from body measurements.

Weight	Crossbreed Males	MSE	R^2^	Purebred Males	MSE	R^2^
W_0_	−2.88 + 0.18(CG)	0.05	0.80	0.24 + 0.09(BL)	0.03	0.89
	−3.13 + 0.13(CG) + 0.33(CP)	0.01	0.96			
W_90_	−16.02 + 1.37(CD)	1.17	0.81	−4.54 + 0.29(CG)	0.35	0.79
	−15.61 + 0.93(CD) + 0.19(BL)	0.80	0.88	−8.02 + 0.20(CG) + 0.71(CW)	0.12	0.93
W_180_	−24.89 + 1.85(CD)	2.00	0.84	−29.29 + 0.76(CG)	0.90	0.96
	−27.88 + 1.38(CD) + 1.03(CW)	1.25	0.91			
W_270_	−37.42 + 0.90(CG)	2.67	0.95	−41.44 + 2.67(CD)	4.49	0.94
	−33.90 + 0.50(CG) + 1.53(CW)	1.85	0.96	−54.27 + 2.03(CD) + 0.55(BL)	2.53	0.97
				−41.92 + 2.25(CD) + 0.62(BL) − 1.66(RL)	1.10	0.99
W_365_	−77.00 + 1.43(CG)	3.60	0.93	−55.58 + 3.15(CD)	1.89	0.98
	−65.93 + 0.88(CG) + 2.18(RW)	1.07	0.98			
W_0–365_	−20.70 + 0.65(CG)	10.23	0.91	−19.52 + 0.63(CG)	11.55	0.90
	−35.36 + 0.43(CG) + 4.37 (CP)	6.14	0.94	−33.61 + 0.41(CG) + 4.45(CP)	6.79	0.95
	−36.54 + 0.12(CG) + 4.67 (CP) + 0.83 (CD)	5.60	0.95			

Note: W_0_: Weight at birth. W_90_: Weight at 90 days (weaning). W_180_: Weight at 180 days. W_270_: Weight at 270 days. W_365_: Weight at 365 days. BL: Body length. CG: Chest girth. CP: Cannon bone perimeter. CD: Chest depth. CW: Chest width. RW: Rump width. RL: Rump length. MSE: Mean square error. R^2^: Coefficient of determination.

**Table 4 animals-13-01978-t004:** Multiple linear regression equations to estimate the live weight of crossbreed and purebred females from body measurements.

Weight	Crossbreed Females	MSE	R^2^	Purebred Females	MSE	R^2^
W_0_	−1.65 + 0.12(WH)	0.03	0.89	−4.49 + 0.22(CG)	0.05	0.91
	−2.45 + 0.11(WH) + 0.26(RW)	0.01	0.96			
W_90_	−20.52 + 2.65(CW)	2.59	0.57	−10.65 + 0.47(BL)	0.73	0.76
W_180_	−28.12 + 1.92(CD)	1.20	0.90	−24.54 + 0.65(CG)	1.56	0.72
W_270_	−26.88 + 0.69(CG)	2.28	0.94	−58.77 + 3.35(CD)	1.03	0.98
				−65.67 + 2.89(CD) + 0.34(BL)	0.61	0.99
				−64.75 + 2.88(CD) + 0.40(BL) − 0.29(RW)	0.28	1.00
W_365_	−24.15 + 0.68(CG)	2.55	0.94	−52.70 + 1.11(CG)	2.01	0.89
				−75.68 + 1.08(CG) + 0.43(WH)	0.91	0.96
W_0–365_	−15.22 + 0.51(CG)	5.46	0.91	−11.01 + 2.04(CW)	7.52	0.89
	−10.95 + 0.68(CG) − 0.28(WH)	4.91	0.92			
	−9.73 + 0.54(CG) − 0.36(WH) + 0.87(CW)	4.42	0.93			

Note: W_0_: Weight at birth. W_90_: Weight at 90 days (weaning). W_180_: Weight at 180 days. W_270_: Weight at 270 days. W_365_: Weight at 365 days. WH: Withers height. RW: Rump width. CW: Chest width. CD: Chest depth. CG: Chest girth. BL: Body length. RW: Rump width. MSE: Mean square error. R^2^: Coefficient of determination.

**Table 5 animals-13-01978-t005:** Live weight prediction equations from the chest girth of Socorro Island Merino lambs.

Exponential	MSE	Pseudo R^2^	Linear	MSE	R^2^
W_0_ = 0.2529 exp^(0.0735 CG)^	0.09	0.71	W_0_ = −3.511 + 0.194(CG)	0.08	0.76
W_90_ = 1.7151 exp^(0.0342 CG)^	1.67	0.59	W_90_ = −9.288 + 0.375(CG)	1.60	0.61
W_180_ = 0.8597 exp^(0.0477 CG)^	2.57	0.80	W_180_ = −24.064 + 0.657(CG)	2.61	0.80
W_270_ = 1.4340 exp^(0.0401 CG)^	5.28	0.90	W_270_ = −40.053 + 0.925(CG)	5.64	0.90
W_365_ = 1.2194 exp^(0.0424 CG)^	4.59	0.92	W_365_ = −47.492 + 1.032(CG)	6.98	0.88
W_0–365_ = 0.9142 exp^(0.0462 CG)^	3.36	0.96	W_0–365_ = −18.540 + 0.598(CG)	10.19	0.89

Note: W_0_: Weight at birth. W_90_: Weight at 90 days (weaning). W_180_: Weight at 180 days. W_270_: Weight at 270 days. W_365_: Weight at 365 days. CG: Chest girth. MSE: Mean square error. Pseudo R^2^: Non-linear coefficient of determination. R^2^: Linear coefficient of determination.

## Data Availability

The data that support the results of this study are available from the corresponding author upon reasonable request.

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
