# Peer review of "Morphological Description and Live Weight Prediction from Body Measurements of Socorro Island Merino Lambs"

_animals, 2023, doi:10.3390/ani13121978_

Round 1
Reviewer 1 Report
Dear Authors,
The manuscript reports on Morphological Description and Live Weight Prediction from Body Measurements of Socorro Island Merino Lambs. The research is conducted on an unknown number of animals and the measurements are also carried out, it is not clear how many there are on each animal. Therefore the structure of the research must be clearly clarified so that each reader can understand the effect of the parameters measured on the live weight.
The study may have an interest in the conservation of these sheep breeds even though conservation is different from livestock production.
In this form it is difficult to assess the value of the study and therefore profound changes have to be made.
Kind regards
English can be improved to be more readable.
Author Response
Dear reviewer, we appreciate your comments and suggestions and we inform you that in accordance with them we have made the following changes:
The initial number of animals included in the study was specified and it was clarified in the discussion why their number subsequently decreased.
New information was included to make the methodology used more understandable.
Various modifications were made to the manuscript to improve its reading.

Reviewer 2 Report
Dear Authors
Animal husbandry is one of the oldest productive endeavor of man, who derived benefits from it many centuries before the historical era. The first diversification of the functional types of livestock took place over the millennia and was more the result of natural selection than of conscious human work. The progressive intensification of livestock breeding was the reason why many breeds or varieties of livestock lost the economic justification for further breeding, became extinct or are now threatened with extinction. This leads to a very negative phenomenon - a reduction in genetic diversity. Indigenous breeds are often perfectly suited to difficult environmental conditions, can be maintained on scarce fodder resources, produce of unique quality, have a centuries-old tradition of production, are long-lived, resistant to disease and show good fertility. These breeds should therefore be preserved and promoted, especially in areas where there is a living tradition of breeding them. The problem addressed in this article to estimate live weight from body measurements of Socorro Island Merino lambs is undoubtedly timely and necessary. The article is written in an interesting way in the individual sections. In the Materials and Methods section, I would introduce larger photos of representatives of the breed in question and, if possible, a photo of the ancestor of this native breed as well. I would complete the Conclusions section, as it is insufficient in its present form.
Regards
Minor editing of English language required.
Author Response
Dear reviewer, we appreciate your comments and suggestions and we inform you that in accordance with them we have made the following changes:
A photograph was included in the document that shows a landscape of Socorro Island in which the deforestation caused by the animals can be seen (it is estimated that before eradicating them there were between 3,500 and 5,000 heads), which was one of the reasons for eradicating them, since The Revillagigedo Archipelago was declared a protected natural reserve. In addition, it shows a photo of some of the sheep captured by the students and researchers of the University of Colima during the rescue expedition from which the lambs included in this study descend. Virtually no photographs exist before these.
Conclusion was modified to more fully reflect the results of the study.

Reviewer 3 Report
The article presents a simple experiment that is well conducted and properly analyzed, but not of interest to the Animals reader. As the authors stated, there are many studies based on body weight estimation from body measurements. Therefore, the article does not make a significant contribution to the literature and does not add anything new.
There are very few animals anyway. Authors are better off having their work published in a local journal.
Author Response
Dear reviewer, we appreciate your comments and suggestions and we inform you that according to them we want to add the following:
It is true that the use of body measurements to estimate the live weight of animals has been discussed previously, however practically all the articles on the subject use simple or multiple linear regression models to generate the prediction equations.
A couple of years ago our working group carried out an extensive search on the use of nonlinear models to estimate live weight of sheep from body measurements and we barely found a dozen publications on the subject.
Derived from the above we write the article: Macedo-Barragán, R., Arredondo-Ruiz, V., Haubi-Segura, C. Castillo-Zamora, P. Comparison of linear and nonlinear models to estimate body weight of Pelibuey ewes from body measurements. Trop Anim Health Prod 53, 56 (2021). https://doi.org/10.1007/s11250-020-02515-z, in which we conclude that non-linear models such as exponential and incomplete gamma (generally used to describe lactation curves) present a better fit than the linear model when chest girth is used as the only estimator. We must point out that since its publication this article has been used as a reference for new investigations in which the same conclusions have been reached, although we must also add that these are still scarce.
In this manuscript that you kindly reviewed, it is demonstrated once again how the exponential model better predicts the weight of the sheep from chest girth than the linear model, which is not unimportant, and we are certain that this finding should be published in a high-impact journal such as Animals and not in a local journal where there is a risk of having little diffusion (in publications from the year 2022 and 2023 the use of linear models is still widely observed).

Round 2
Reviewer 1 Report
Dear Mr Devon Liu
Manager Office of Animals
The manuscript was improved and could be published in this form. However, the inclusion of who the breed is and how widespread it is would be appropriate for the completeness of the study and to understand how useful this study is.
Kind regards
Author Response
Dear reviewer, a paragraph was included at the end of the discussion indicating some characteristics for which this breed should be preserved.
Reviewer 3 Report
My thoughts on this article remain the same. I think the article is pretty mundane for Animals readers. But I respect the opinions of other referees. If they make an opinion on the acceptance of the article, it's fine with me.
Author Response
Dear reviewer, we appreciate your comments, it is these that allow us to improve the quality of future manuscripts.